# CREW: Facilitating Human-AI Teaming Research

**Lingyu Zhang**                                           *lingyu.zhang@duke.edu*
*Duke University*

**Zhengran Ji**                                            *zhengran.ji@duke.edu*
*Duke University*

**Boyuan Chen**                                            *boyuan.chen@duke.edu*
*Duke University*

**Project Website:** `http://www.generalroboticslab.com/CREW`

**Reviewed on OpenReview:** `https://openreview.net/forum?id=ZRXwHRXm8i`

## Abstract

With the increasing deployment of artificial intelligence (AI) technologies, the potential of humans working with AI agents has been growing at a great speed. Human-AI teaming is an important paradigm for studying various aspects when humans and AI agents work together. The unique aspect of Human-AI teaming research is the need to jointly study humans and AI agents, demanding multidisciplinary research efforts from machine learning to human-computer interaction, robotics, cognitive science, neuroscience, psychology, social science, and complex systems. However, existing platforms for Human-AI teaming research are limited, often supporting oversimplified scenarios and a single task, or specifically focusing on either human-teaming research or multi-agent AI algorithms. We introduce **CREW**, a platform to facilitate Human-AI teaming research in real-time decision-making scenarios and engage collaborations from multiple scientific disciplines, with a strong emphasis on human involvement. It includes pre-built tasks for cognitive studies and Human-AI teaming with expandable potentials from our modular design. Following conventional cognitive neuroscience research, CREW also supports multimodal human physiological signal recording for behavior analysis. Moreover, CREW benchmarks real-time human-guided reinforcement learning agents using state-of-the-art algorithms and well-tuned baselines. With CREW, we were able to conduct 50 human subject studies within a week to verify the effectiveness of our benchmark.

## 1 Introduction

Over the past decade, significant progress in machine learning has increased the potential and necessity for humans to collaborate and interact with Artificial Intelligence (AI) agents. Human-AI teaming has emerged as a research paradigm to explore the dynamic interactions and collaborative processes between humans and AI. By leveraging the complementary strength of both humans and AI, advancements can significantly enhance the overall team performance.

Unlike traditional AI research, which typically focuses on machine learning algorithms in isolation, Human-AI teaming requires a multidisciplinary approach to incorporate insights from various scientific domains. Numerous studies have examined human-human teaming Klimoski & Mohammed (1994) with cognitive science, neuroscience, and psychology. Machine learning and robotics communities have extensively researched multi-agent machine learning Zhang et al. (2021), while team dynamics Salas et al. (2018) has been explored in complex systems, social science, and network science. Despite the importance and potential of this research

paradigm, there is still a lack of a comprehensive and unified platform to benefit research on joint efforts across disciplines and scalable hypothesis verification.

Developing a comprehensive platform for Human-AI teaming research presents several unique challenges. Firstly, the platform needs to support diverse tasks with varying complexities with easy extensions for new tasks or modifications. While reinforcement learning research platforms Towers et al. (2023) have widely adopted this practice, current Human-AI teaming platforms remain limited to single tasks Carroll et al. (2019); Vinyals et al. (2017). Secondly, enabling real-time communication through various modalities between multiple humans and heterogeneous AI policies is essential for effective collaboration. However, existing solutions typically support human feedback only through replaying offline trajectories and do not implement real-time feedback mechanisms. Understanding how to build AI that can team with, learn from, be guided by, align with, and augment humans is as crucial as modeling human behavior during interactions with AI. Therefore, the platform must provide interfaces for recording human physiological data alongside agent data, tailored for cognitive science and neuroscience studies. Furthermore, current studies often involve small participant groups, making it difficult to derive generalizable conclusions. Lastly, the absence of a unified platform has limited fully open-sourced studies and the establishment of appropriate benchmarks.

We present CREW, a platform designed to facilitate Human-AI teaming research aiming to address the above challenges. We develop CREW around key principles such as extensible and open environment design, real-time communication support, hybrid Human-AI teaming modes, parallel sessions for scalable experiments, and comprehensive human and agent data collection. Additionally, CREW incorporates highly modular algorithm components compatible with practices in the machine learning community. We demonstrate CREW's potential by benchmarking real-time human-guided reinforcement learning (RL) algorithms alongside various RL baselines. With CREW, we were able to conduct 50 human subject studies within a week. Moreover, CREW includes a set of cognitive tests to explore how individual differences among humans impact their effectiveness in training AI agents. To our knowledge, CREW is the first platform to unify the desired features for Human-AI teaming research across multiple disciplines. We aim for CREW to serve as an infrastructural foundation for multidisciplinary, reproducible, and scalable Human-AI teaming research.

## 2 Related Work

**Human-AI Teaming Research** Extensive research has explored Human-AI teaming across various domains. Machine learning studies have developed algorithms to leverage human expertise to improve the accuracy Lertvittayakumjorn et al. (2020), robustness Bao et al. (2018); Ziegler et al. (2019), and interpretability Lertvittayakumjorn et al. (2020); Bao et al. (2018); Zhou & Chen (2019) of models. Integrating human feedback can not only improve performance Ibarz et al. (2018); Christiano et al. (2017) but also align the models with human preference Lertvittayakumjorn et al. (2020); Ouyang et al. (2022); Ziegler et al. (2019). Human-computer interaction research has created interfaces Weisz et al. (2021); Neerincx et al. (2018) and workflows Bau et al. (2020) that enhance collaboration between humans and AI, combining their strengths to achieve superior performance. Ethical research focuses on understanding and mitigating the societal Hemmer et al. (2023); Chowdhury et al. (2022), ethical Ezer et al. (2019); Yin et al. (2019); Pflanzer et al. (2023), and technical Green & Chen (2019); Stahl et al. (2021) challenges of the rapid advancement and wide adoption of AI. Many fields, including neuroscience, healthcare, robotics, transportation, education, security, and accessibility, have shown growing interest Nourbakhsh et al. (2005); Henry et al. (2022); Atakishiyev et al. (2021); Nwana (1990); Pazho et al. (2023); Kumar & Jain (2022) in Human-AI teaming. Overall, the broad spectrum of interests highlights the need for multidisciplinary collaboration to drive further advancements.

**Human-AI Teaming Platform** While significant progress has been made in Human-AI teaming research, there remains an absence of a comprehensive research platform. Overcooked Environment Carroll et al. (2019) is a simplified version of the original popular game to challenge human agents and AI agents in tasks that require close coordination and strategic teamwork. StarCraft II Learning Environment (SC2LE) Vinyals et al. (2017) supports adversarial settings to allow Human-AI interaction and learning from human demonstrations. Rapid Integration and Development Environment(RIDE) USC Institute for Creative Technologies (2024) focuses on defense-related scenarios, emphasizing rapid development and integration of AI systems for operational purposes. In addition to real-time decision-making tasks, previous research has developed

| Platform | Extensible envs | Real-time interaction | Multiplayer | Multimodal feedback | Physiological data | Fully open-sourced |
|---|---|---|---|---|---|---|
| CREW | ✔ | ✔ | No Limit | ✔ | ✔ | ✔ |
| Overcooked Carroll et al. (2019) | | ✔ | 1+ 1 | | | ✔ |
| SC2LE Vinyals et al. (2017) | | ✔ | 1 v 1 | | | |
| RIDE USC Institute for Creative Technologies (2024) | ✔ | ✔ | No Limit | | | |
| RLHF-Blender Metz et al. (2023) | | ✔* | | ✔ | | ✔ |
| Uni-RLHF Yuan et al. (2024) | | ✔* | | ✔ | | ✔ |

Table. 1: Human-AI teaming platforms. *Instead of real-time feedback training, RLHF-Blender and Uni-RLHF support online training where humans are presented with data from the replay buffer instead of the current experience.

platforms Freire et al. (2020); Metz et al. (2023); Yuan et al. (2024) that focus on offline preference or rating settings where humans can provide offline evaluations or corrections with imitation learning or reinforcement learning.

Overall, existing platforms have more than one of the following limitations that constrain Human-AI teaming research as summarized in Table. 1. Most environments only support one type of task that can be difficult to extend, and the interactions between humans and machine learning agents are limited to mouse and keyboard operations. Moreover, most of the environments only support collecting game data, such as state, action, or reward focusing on the machine learning algorithm development, but none of them support the collection and analysis of human physiological data (eye movement, pupillometry, brain activity, heart impulse, or natural language) to understand human cognitive states and different effects on human subjects, as often required in the cognitive studies involved in Human-AI teaming Thakur et al.; Dikker et al. (2017); Heinisch et al. (2024); Gordon et al. (2020); Koorathota et al. (2023); Xu et al. (2021). Additionally, the scale of collaboration has been limited to two agents in a collaborative or competitive setting. Notably, most existing Human-AI teaming studies have only been evaluated on a very small number of subjects or among the authors ($N < 10$, mostly $N < 5$), which greatly limits our understanding of the state-of-the-art performance. Furthermore, common previous multi-agent environments such as OvercookedCarroll et al. (2019) and Starcraft Vinyals et al. (2017) are closed-sourced, making it unrealistic to build additional features on top of them. This also makes it impossible for researchers to modify the tasks to their own needs. There remains a large gap between the existing platforms and the desired environment to facilitate future research.

**Real-Time Human-Guided Learning** Most real-world decision-making processes require rapid decisions and adaptation in real time. Therefore, real-time human-guided learning is an essential topic in Human-AI teaming research. Previous research has focused on algorithm design to integrate real-time human feedback into policy training. TAMER and Deep TAMER Knox & Stone (2009); Warnell et al. (2018) learn to predict human discrete feedback and utilize the feedback model as the one-step myopic value function for policy learning. COACH and DeepCOACH Arumugam et al. (2019); MacGlashan et al. (2016) addressed the inconsistency of human feedback by modeling it as the advantage function under an actor-critic framework. Recent progress has further refined these algorithms, addressing various challenges in human feedback integration, such as different feedback modalities feedback stochasticity Arakawa et al. (2018), and continuous action spaces Sheidlower et al. (2022). Other works have explored indirectly inferring human objectives from feedback or preference Huang et al. (2023); Xiao et al. (2020); Christiano et al. (2017). CREW provides an extensive environment for real-time human-guided learning with online training and feedback integration, human physiological data collection, and parallel distributed experiment support.

## 3 CREW: Design and Components

### 3.1 Platform Vision and Design Philosophy

We design CREW to facilitate Human-AI teaming research. Our vision is to construct a unified platform for researchers from diverse backgrounds, allowing them to join forces from human-AI interaction, machine learning algorithms, workflow design, as well as human analysis and training. To achieve this, CREW incorporates the following capabilities, as illustrated in Figure. 1.

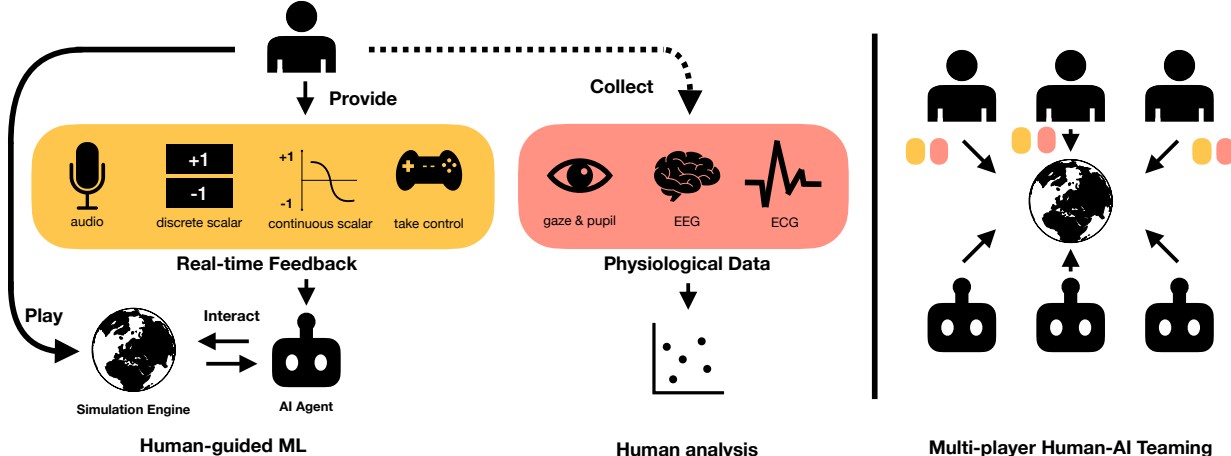

Figure. 1: **CREW** is a platform to facilitate Human-AI teaming research. CREW is designed under the vision of multidisciplinary collaboration research from both human and AI science. CREW allows real-time interaction among multi-human and multi-agents while enabling extensive data collection on both AI agents and human agents.

**Extensible and open environment design.** CREW is fully open-sourced. Our key contribution is building the infrastructure to allow the development of scalable environments with supported Human-AI research features that scale up to very complex settings. We implemented a set of built-in example tasks described in Section 3.2 for rapid development. Many additional features and game logic can be added without modifying complex infrastructure-level components.

**Real-time communication.** While some Human-AI interaction tasks, such as human preference-based fine-tuning, can be performed offline, many applications require online real-time interaction. Whether it is training decision-making models with real-time human guidance or general human-AI collaboration tasks, the ability to convey messages with minimum delay is essential. Synchronizing data flow between human interfaces, AI algorithms, and simulation engines necessitates the establishment of a real-time communication channel.

**Hybrid Human-AI teaming support.** Teaming is an essential aspect of our daily jobs. Our vision extends this concept to Human-AI teaming, where both humans and AI operate in teams. Increasing interest in the organization, dynamics, workflow, and trust in multi-human and multi-AI teams highlights the need for a platform capable of distributing and synchronizing tasks, states, and interactions across multiple environment instances and even across physical locations.

**Parallel sessions support.** A key bottleneck for human-involved AI research is the requirement to conduct experiments with dozens or hundreds of human subjects to obtain trustworthy and reliable conclusions. Such a process can be tedious and time-consuming. To enhance efficiency and scalability, CREW supports multiple independent parallel sessions of the same setting, unconstrained by geographical locations, to obtain the "crowd-sourcing" effects of large-scale experiments. This capability enables experimenters to collectively share experimental data and results.

**Comprehensive human data collection.** Though human plays an important role in Human-AI teaming, our understanding of human behaviors remains limited and under-explored in existing studies. Therefore, CREW offers interfaces to simultaneously collect multi-modal human data, ranging from active instructions and feedback to passive physiological signals.

**ML community-friendly algorithm design.** The choice of programming language and libraries should align with the customs and preferences of the ML community. The system design should be modular to allow seamless transitions between tasks and algorithms.

## 3.2 Environments

**Tasks** We select Unity as the simulation engine for CREW due to its popularity in game design and AI research to allow extensible and open environment design. Unity, as used by many popular commercial games as has a high ceiling for the complexity of game development. It also has a large user community which makes customized task development more accessible. Adding new components to an environment can be done through simple drag-and-drop (on the left of Figure. 2 ).

We have implemented four challenging tasks as examples. Multi-player tasks are designed for multi-agent and multi-human teaming research, and single-player tasks are designed for human-guided AI agent learning studies. For each task, we provide both visual and structured state input options. The detailed settings are summarized in Table. 2.

Bowling is a modified version of Atari bowling where each round consists of 10 rolls and the score for each roll is the number of pins hit. Bowling is designed to have the simplest dynamics among our tasks to serve as a rapid test for training a single agent. Find Treasure (Figure. 3A) is a single-player embodied navigation task where the agent's goal is to explore a maze and reach the treasure with randomized initial and goal locations. 1v1 Hide-and-Seek Chen et al. (2020; 2021a) (Figure. 3B) is a one-on-one hide-and-seek task where the seeker learns to explore the maze and catch a moving hider that follows a heuristic policy for obstacle avoidance, and run away from the seeker within line of sight. We introduce this task as an adversarial competition setting. NvN Hide-and-Seek (Figure. 3C) is a multiplayer version where multiple seekers and hiders can coordinate, collaborate, and compete. The hiders and seekers can either be controlled by humans or heterogeneous AI policies.

Table. 2: Task Specifications

| Tasks | Visual Observation | State Observation | Action Space | Reward |
|---|---|---|---|---|
| Bowling | grayscale | ball pos, pin pos, pin status | release pos, length before steer, steer direction | # pins hit |
| Find Treasure | rgb | agent pos, treasure pos | next loc x, next loc y | -1 / step, +10 treasure found |
| 1v1 Hide-and-Seek | rgb | seeker pos, hider pos | next loc x, next loc y | -1 / step, +10 hider caught |
| NvN Hide-and-Seek | rgb | seekers pos, hiders pos | next loc x, next loc y | user define |

**Customizing new tasks or easily extend current tasks.** Researchers can build tasks of much higher flexibility and complexity in CREW than existing platforms due to our efforts in setting distributed multi-agent systems mixing human and AI agents. We chose the set of tasks above only for benchmarking purposes for human-guided reinforcement learning experiments, and it is far below the complexity limit of game design that CREW can support. As shown in Figure. 2 on the right side, we can easily scale up the difficulty of a Hide-and-Seek game to a more complex map. Another example is a recent work Ji et al. (2024) that shows the usage of CREW for human-guided multi-agent Hide-and-Seek.

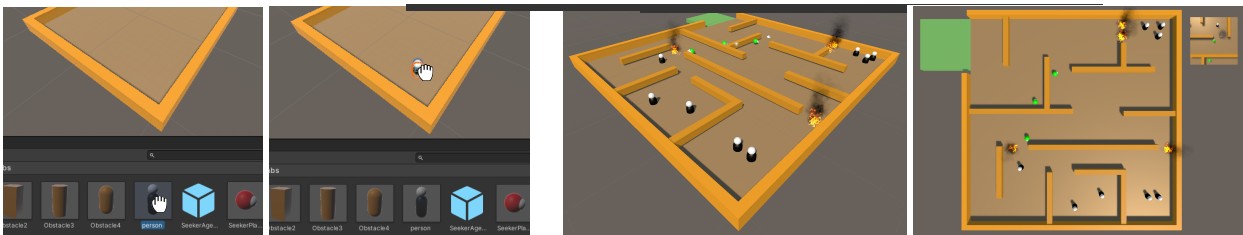

Figure. 2: Left: Example of adding objects to an environment in CREW through drag-and-drop. Right: Example of scaling up the complexity of hide-and-seek to a search-and-rescue task.

**Visual Observations** Different visual observations create various challenges in visual embodiment learning for both humans Vander Heyden et al. (2017); Fishbein et al. (1972) and AI agents Bandini & Zariffa (2020); Tarun et al. (2019). We provide various camera configurations to support perceptual-motor research. As shown in Figure.3, for all our navigation and competitive tasks, we implemented visual observations from multiple views for the users' selection: a top-down fully observable view, a top-down accumulated partially

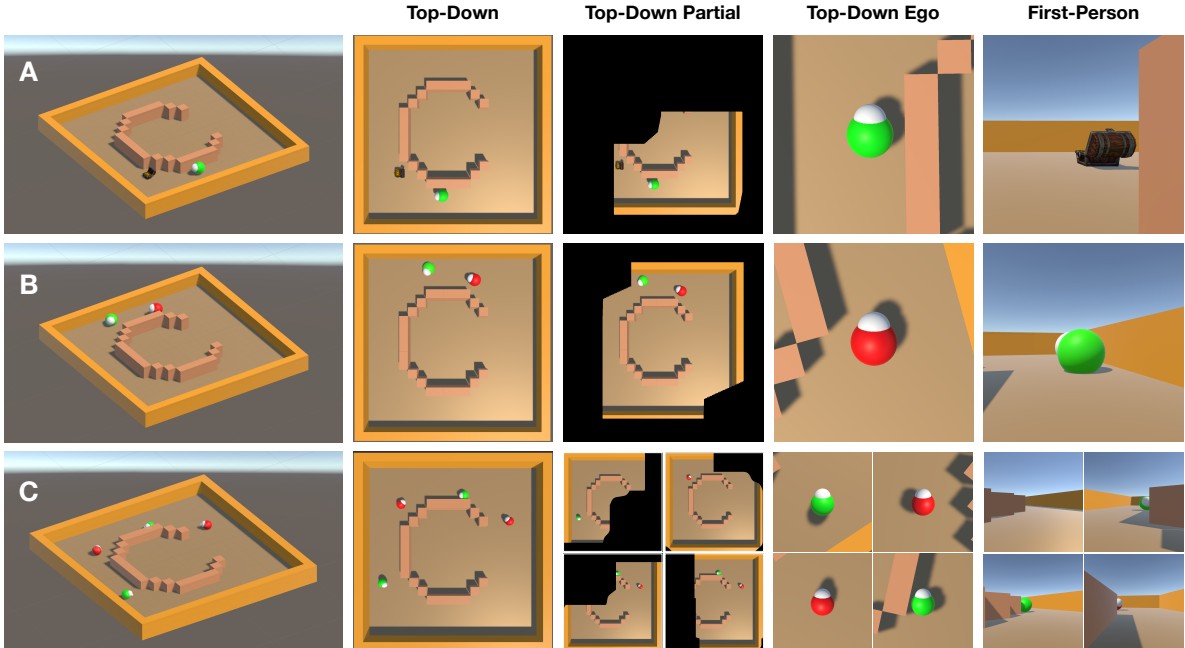

Figure. 3: CREW supports multiple tasks from single agent (A: Find Treasure) to multi-agent competitive setting (B: 1v1 Hide-and-Seek), and multi-agent collaborative and competitive setting (C: NvN Hide-and-Seek). We also show different camera views supported by CREW for perceptual-motor research.

observable view similar to SLAM in robotics Smith & Cheeseman (1986), a top-down egocentric view, and a first-person view.

**Procedural Generation** Learning robust, generalizable, and scalable AI agents requires diverse training environments. Procedural generation allows for the creation of a wide range of environment patterns and terrain types. We provide randomized maze patterns where the grids are guaranteed to be connected (Figure. 4A) and procedural-generated terrains for more complex simulations as in Figure. 4B.

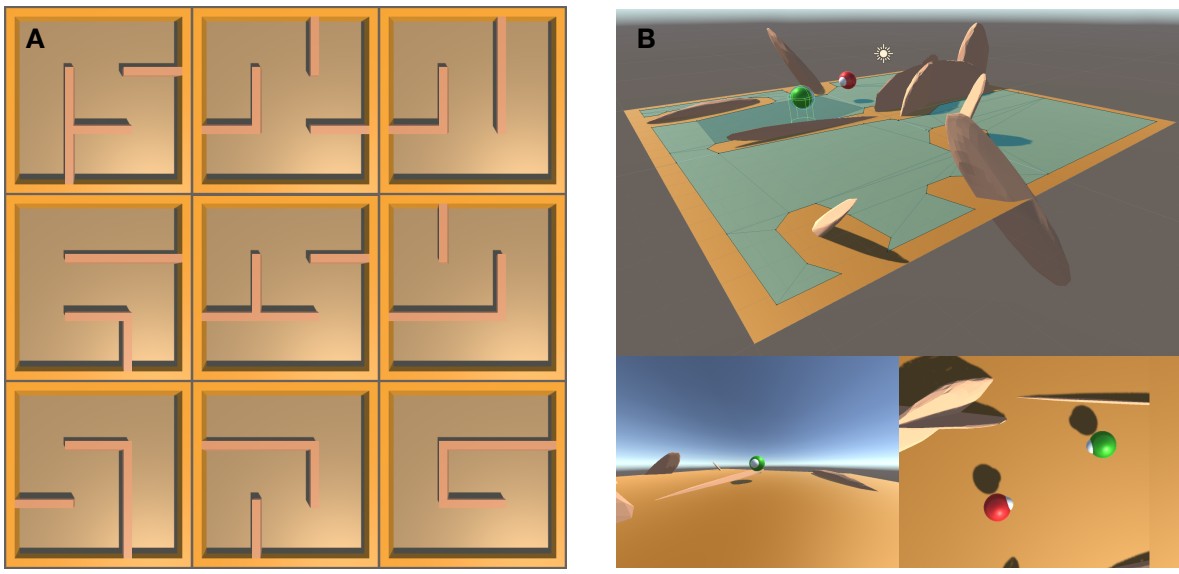

Figure. 4: Environment generation in CREW. (A) Randomized maze. (B) Procedure generated terrains.

### 3.3 Human and Agent Role Assignment

Humans and AI agents often have complementary strengths. For example, humans are generally better at exploring and adapting to new situations, while AI agents are good at repetitive exploitation and precise calculation. Naturally, a team consists of humans and AI agents should have various roles to be effective. Different roles can also be assigned within AI agents to study multi-agent machine learning with heterogeneous policies. To facilitate these experiments, we provide the role assignment feature in CREW.

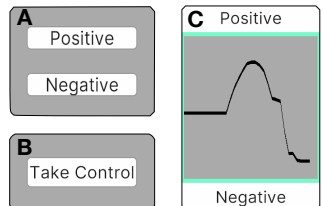

"Player" allows humans to directly control an agent. "Viewer" allows humans to observe and provide feedback signals as guidance to an AI agent. "Server" role allows humans to host the entire task by monitoring it without direct participation. "AI Agent" assigns different learning policies to each agent.

**Human Feedback Interface** We provide a user interface to allow the Viewer role to provide direct feedback to AI agents shown in Figure. 5. Scalar feedback is the most common feedback type used in human-guided RL Knox & Stone (2009); Warnell et al. (2018); Xiao et al. (2020). Conventionally, a human chooses to provide a positive or negative rating to the state-action pair of an agent

Figure. 5: (A) Discrete scalar feedback. (B) Option to take control of the agent and teleoperate. (C) Continuous scalar feedback: the human can hover the mouse over this window to provide per-step feedback.

policy. Additionally, CREW offers an interface that allows humans to provide feedback that is continuous in value and time, enabling granular feedback information. Moreover, our interface also allows humans to directly take control over agents and perform teleoperation.

### 3.4 Multiplayer and Parallel Sessions

Enabling multi-human multi-agent sessions requires robust networking solutions (Figure. 6). We use Unity Netcode net for game state synchronization, and Nakama nak as the networking server. In CREW, a server instance hosts the task, runs the simulation, and handles agent policy training, which can be executed on a powerful headless GPU server. Human participants can connect via client instances on less powerful machines, which display synchronized game states and collect human input. CREW is cross-platform, allowing participation from Linux, Windows or MacOS machines.

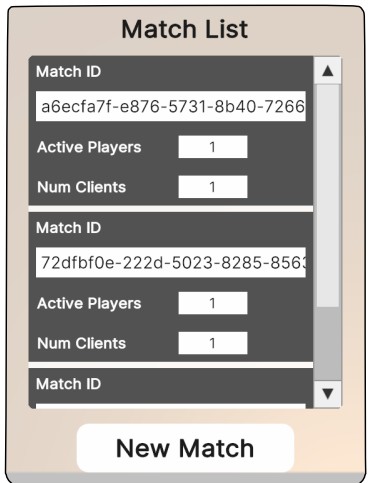
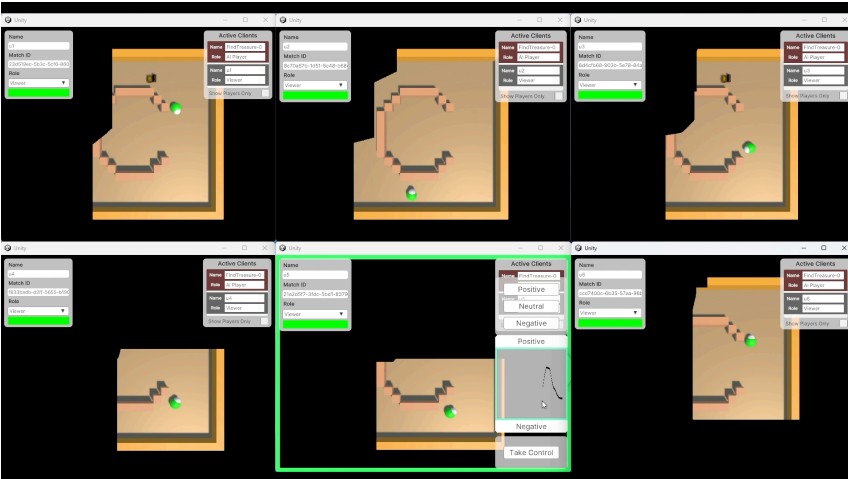

Figure. 6: In CREW, it is possible to connect and join multiple instances of tasks. It is as simple as typing in an IP address and selecting the task one wants to join.

### 3.5 Human and Agent Data Collection

Data collection is at the core of Human-AI teaming research. CREW includes a pipeline for thorough data collection on both the human side and AI agent side.

**Human data** Prior research in Human-AI teaming and human-human teaming has leveraged physiological features in various ways. Thakur et al. uses predicting human eye gaze as an auxiliary task for guiding imitation learning. Dikker et al. (2017) conducts brain synchrony with EEG in a classroom multi-human teaming setup. Qin et al. (2022) leverages pupil dynamics to predict the team performance in a multi-human virtual reality task. Heinisch et al. (2024) uses physiological data for human-robot interaction with service robots. Gordon et al. (2020) explored physiological synchrony in teaming. Koorathota et al. (2023) used gaze data to enhance vision transformers. Xu et al. (2021) uses EEG data as implicit human feedback for accelerating RL. We implemented the relevant features to support future research along these lines. Besides the feedback interfaces that collect feedback signals of multiple modalities and teleoperation actions, we also provide interfaces for collecting audio, eye gaze, pupillometry, electroencephalogram (EEG), and electrocardiogram (ECG) physiological responses as in Figure. 7 with time synchronizations from each machine with Lab Streaming Layer Kothe et al. (2024).

**Agent data** including the policy weights, observations, actions, rewards, feedback received, and loss values at every time step can all be saved for further analysis. Users also have the option to enable experiment monitoring and logging by Weights & Biases Biewald (2020). As all of our tasks include vision-based settings, we also provide implementations of a set of vision encoder architectures.

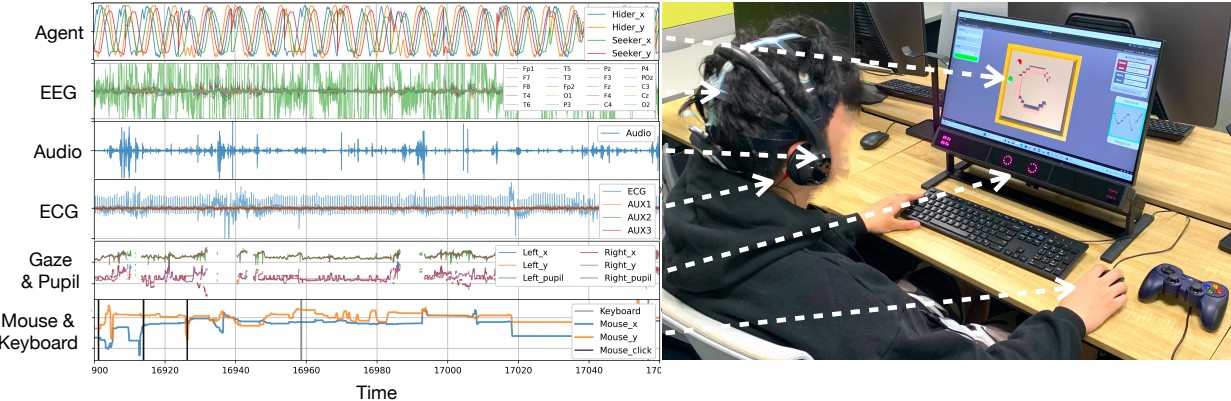

Figure. 7: Data collection in CREW. All human and agent data, including game states, feedback, mouse and keyboard, audio, gaze and pupil data, EEG and ECG data are all streamed and synchronized through Lab Streaming Layer.

### 3.6 Designing Algorithms

Algorithms research is crucial for Human-AI teaming. We designed the algorithm component of CREW to be extensible and accessible to the ML community. Algorithms are implemented in Python with PyTorch Paszke et al. (2019) and TorchRL Bou et al. (2023) which is part of the PyTorch ecosystem, making it easy for researchers to design and deploy new algorithms. We provide implementations of state-of-the-art human-guided machine learning algorithms and strong reinforcement learning baselines. The API for communication between a Unity instance and Python algorithm is implemented with MLAgents Juliani et al. (2018). All environments follow a uniform communication protocol for the observation and action data across tasks. Our modular design allows smooth switching between tasks and algorithms. Real human experiments are time-consuming and costly. To ease algorithm debugging, we implemented simulated human feedback providers as surrogates for real humans for complex tasks. These simulated feedback uses heuristics based on prior task knowledge and is not available in novel tasks in the real world; hence, they should be used solely as debugging tools, not as replacements for real human evaluations.

Table. 3: Cognitive Tests Specifications

| Tests | Rule | Score Metric |
|---|---|---|
| Eye Alignment (eye) Scharre (2014) | Align a ball on the left side of the screen with a target on the right side as accurately as possible within five seconds. | Negative average of the distances between the ball and target's horizontal positions across trials. |
| Reflex (reflex) Boes (2014) | Click the screen as quickly as possible when it changes from yellow to green. | Negative average of the response times. |
| Theory of Behavior (theory) Chen et al. (2021b) | Observe a red ball moving in an unknown but fixed pattern for 5s. When the ball pauses, predict the ball's position one second after it resumes moving. | Negative average of the distances between the ball's actual and predicted positions across trials. |
| Mental Rotation (rotation) Shepard & Metzler (1971) | Identify the piece among three similar pieces that cannot be rotated to match the target piece. | Accuracy of the subject's identifications across all trials. |
| Mental Fitting (fitness) Shepard & Metzler (1971) | Identify the only piece among three similar pieces that can fit with the target piece. | Accuracy of the subject's identifications across all trials. |
| Spatial Mapping (spatial) Berkowitz et al. (2021) | A video of an agent navigating a maze with a restricted field of view is presented. Identify the maze from a selection of three similar mazes. | Accuracy of the subject's identifications across all trials. |

### 3.7 Modular Pipeline Design for Quantifying Human Characteristics

Individual differences among humans can significantly affect their teaming with AI agents. To support research along this line, CREW supports a set of cognitive tests to quantify these differences shown in Table. 3. We provide a modular and convenient pipeline (Figure. 8) for executing cognitive tests and Human-AI experiments. The framework integrates various media files (e.g., instruction videos or pictures), inter-trial intervals, executable Python scripts, and Unity builds, ensuring a smooth and effective workflow. The pipeline requires minimal effort from researchers during proctoring, as a single click initiates the sequential execution of experiments. The pipeline allows restart from interruption points.

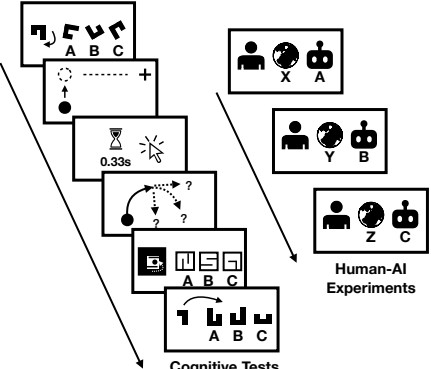

Figure. 8: Modular pipeline for experiment setup. Experimenters can freely select and organize the order of cognitive tests and Human-AI experiments with CREW's deployment pipeline.

## 4 Benchmarking Study

As an example of running experiments with CREW, we benchmark a state-of-the-art real-time human-guided RL framework, Deep TAMER Warnell et al. (2018), along with strong RL baselines. In the original Deep TAMER, the framework was only tested on Atari Bowling with 9 human subjects. With CREW, this is the first time it is possible to systematically conduct human-guided RL benchmarking across multiple environments on a larger population. We summarize our findings as well as insights on the scalability of the framework in this section. We also discuss the relationship between human characteristics and guided agent performance.

### 4.1 Experiment Setup

**Tasks** We selected 3 single-player games: Bowling, Find Treasure, and 1v1 Hide-and-Seek for this benchmark. For Find Treasure and Hide-and-Seek, each episode has a time limit of 15 seconds. All algorithms directly learn from visual inputs with the top-down accumulated partially observable view.

**Human Trainers** We recruited 50 human subjects for the experiments under the approval of the Institutional Review Board. We highlight that our experiment is the largest-scale study so far on real-time human-guided AI training. For every human subject, the experiment time is approximately 40 minutes without interruptions. The experiment starts with cognitive tests (10 minutes) and is followed by the human guiding the agent using the c-Deep TAMER (which will be discussed in the next section) framework (30 minutes). The order of the cognitive tests is Eye Alignment, Reflex, Theory of Behavior, Mental Rotation, Mental Fitting, and Spatial Mapping. There are detailed instruction videos for each test before the test starts. As for the human guiding agent component, each human subject guides a single agent to play 3 tasks for a total of 30 minutes(5

minutes for Bowling, 10 minutes for Find Treasure, and 10 minutes for 1v1 Hide-and-Seek). Before each task, there is a detailed instruction video that describes the rule of the task and how human subjects can give feedback to the agent. Benefiting from CREW's unique feature to host experiments with parallel sessions, we were able to conduct all 50 experiments within one week. Prior to guiding the agents, the participants were asked to complete all our cognitive tests in Table. 3.

**Deep TAMER** Warnell et al. (2018) is a well-established baseline that leverages human feedback as myopic value functions. During training, a human trainer provides positive or negative discrete feedback based on the agent's behavior. This feedback is assigned to relevant state-action pairs through a credit assignment window. A neural network is trained to predict human feedback given state-action pairs, and the policy chooses actions that maximize this predicted feedback. Originally, Deep TAMER was limited to discrete actions. We extend it to continuous action spaces using an actor-critic framework, termed as c-Deep TAMER, specified in Alg 1. For bowling, the agent is trained for 5 minutes, and for 10 minutes in Find Treasure and Hide-and-Seek.

---

**Algorithm 1** The c-Deep TAMER algorithm.

---

**Require:** initial policy $A$ parameters $\theta$, human feedback model $H$ parameters $\phi$, empty replay buffer $\mathcal{D}$, step size $\eta$, buffer update interval $b$, credit assignment window $w$

**Init:** $j = 0$, $k = 0$

1: **for** $i = 1, 2, \ldots$ **do**
2:     **observe** state $\mathbf{s}_i$, time $t_i$
3:     **execute** action $a_i = \text{clip}(A_\theta(\mathbf{s}_i) + \epsilon, a_{Low}, a_{High})$, where $\epsilon \sim \mathcal{N}$
4:     $\mathbf{x}_i = (\mathbf{s}_i, \mathbf{a}_i, t_i, t_{i+1})$
5:     **if** new feedback $\mathbf{y}$ **then**
6:         $j = j + 1$
7:         $\mathbf{y}_j = \mathbf{y}$
8:         $\mathcal{D}_j = \left\{ (\mathbf{x}, \mathbf{y}_j) \mid w(\mathbf{x}, \mathbf{y}_j) \neq 0 \right\}$
9:         $\mathcal{D} = \mathcal{D} \cup \mathcal{D}_j$
10:        **update** $H$ by one step of gradient descent using

$$\nabla_\phi \frac{1}{|\mathcal{D}_j|} \sum_{(\mathbf{x}, \mathbf{y}) \in \mathcal{D}_j} ||H_\phi(s_i, a_i) - \mathbf{y}||_2$$

11:        **update** $A$ by one step of gradient ascent using

$$\nabla_\theta \frac{1}{|\mathcal{D}_j|} \sum_{(\mathbf{x}, \mathbf{y}) \in \mathcal{D}_j} H_\phi(s_i, A_\theta(s_i))$$

12:        **update** target networks
13:        $k = k + 1$
14:     **end if**
15:     **if** $\text{mod}(i,b)==0$ and $\mathcal{D} \neq \emptyset$ **then**
16:        Randomly sample a batch of transitions, $B = \{(s, a, y)\}$ from $\mathcal{D}$
17:        **update** $H$ by one step of gradient descent using

$$\nabla_\phi \frac{1}{|B|} \sum_{(\mathbf{x}, \mathbf{y}) \in B} ||H_\phi(s_i, a_i) - \mathbf{y}||_2$$

18:        **update** $A$ by one step of gradient ascent using

$$\nabla_\theta \frac{1}{|B|} \sum_{(\mathbf{x}, \mathbf{y}) \in B} H_\phi(s_i, A_\theta(s_i))$$

19:        **update** target networks
20:        $k = k + 1$
21:     **end if**
22: **end for**

---

**Baseline RL** We selected two state-of-the-art RL algorithms: Deep Deterministic Policy Gradient (DDPG) Lillicrap et al. (2015) and Soft Actor-Critic (SAC) Haarnoja et al. (2018).

**Heuristic feedback** We also evaluate simulated feedback-guided RL. We simply add the feedback signals as additional dense rewards to the environment reward. DDPG is selected as the backbone RL algorithm as the state-of-the-art transitioned from SAC to DDPG Yarats et al. (2021) in visual control tasks. The hyperparameters for the heuristic baseline is set to the same as DDPG baseline.

**Evaluation** We checkpoint model weights every 2 minutes and evaluate on unseen test environments. For bowling, every checkpoint is evaluated for 1 game (10 rolls). For Find Treasure and 1v1 Hide-and-Seek, the checkpoints are evaluated for 100 episodes.

**Processing Inputs** All tasks in our experiments directly learn from pixels where humans and AI agents share the same input modality. The frames rendered from the environment are first resized to $100{\times}100$ pixels. For Find Treasure and 1v1 Hide-and-Seek, we stack the past 3 frames as input to include the history information. We then apply a simple random shift to the frame stack, as it has been shown to be an effective augmentation strategy for visual reinforcement learning. The shifting factors along the height and width are uniformly sampled from [0, 0.08].

**Model architecture** We use a 3-layer CNN as the vision encoder, each having 64 channels and followed by batch normalization and ReLU activation function. All actor-critic frameworks use a 3-layer MLP with 256 neurons in each layer for both the actor and critic network.

**Computational Resources** All human subject experiments were conducted on desktops with one NVIDIA RTX 4080 GPU. All evaluations were run on a headless server with $8 \times$ NVIDIA RTX A6000 and NVIDIA RTX 3090 Ti. We note that we used larger GPU servers due to the need for parallel evaluation resulting from the large scale of our study, but this is not a requirement to run CREW.

### 4.2 Results

**Agent training performance** We hypothesize that subjects with higher cognitive tests can lead to higher-performing agents. Therefore, we show the agent performances guided by the 15 subjects who scored the highest in our cognitive tests side by side with the performance of all 50 subjects in Figure. 9. As shown in the results, the agents guided by the top 15 subjects exhibit higher performance than the overall average. This is most prominent in Find Treasure, where c-Deep TAMER (blue curve) is significantly higher in the Top 15 plot than the All 50 plot. As shown in Figure. 10, the correlation between cognitive test scores and guided AI performance is positive for all tasks, where Find Treasure had a strong statistical significance (**, $p < 0.01$). We do not draw a general conclusion from these early explorations. We hope to provide a useful metric and experiment to study individual differences for future work, exemplified by our benchmarking results. We highlight our concurrent work Zhang et al. (2024) to use them in human-guided RL evaluation.

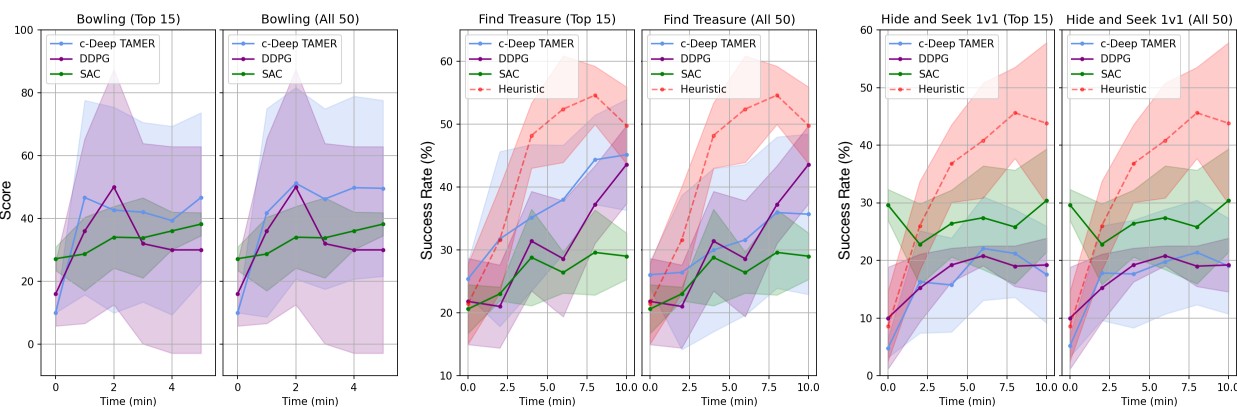

Figure. 9: The average result of the human subjects with the top 15 cognitive test scores and the average result across all 50 subjects. We also find that c-Deep TAMER has difficulties scaling to more complex tasks.

In particular for the top 15 subjects, on the simple bowling task, c-Deep TAMER surpasses RL baselines by an average of 10 scores given the same training time. On Find Treasure, heuristic feedback achieved the highest performance as expected, showing the upper bound performance with accurate and non-delayed dense rewards. c-Deep TAMER also demonstrated strong performance with faster learning trends than RL baselines. On 1v1 Hide-and-Seek, c-Deep TAMER performed similarly to RL baselines, suggesting that c-Deep TAMER has difficulty scaling to tasks with higher complexity. Similar conclusions still hold for all 50 subjects. We also include a comparison between the top and bottom 25 cognitive test scorers in Appendix 6.5.

**Analysis of Individual Differences** Due to the cognitive test feature and emphasis on Human-AI teaming, we can deepen our understanding of how individual human differences can affect the performance of human-guided agents. We calculated the correlation between human subjects' cognitive test scores and c-Deep TAMER training results in Figure. 10. The cognitive test scores are normalized by the mean and variance over the subjects through z-score. Following Tukey (1977), we remove outliers with scores $1.5\times$ Interquartile Range above the third quartile or below the first quartile are removed. We found that non-outlier subjects performed equally well on the spatial mapping test so we do not include this for correlation analysis. The heatmap shows the sign of correlation coefficients from linear regression and the statistical significance (*).

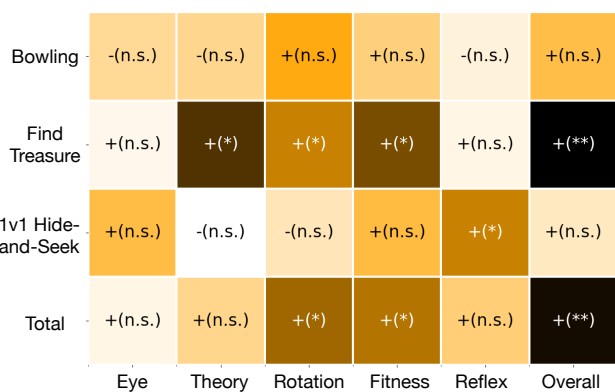

Figure. 10: Correlation between cognitive test scores and c-Deep TAMER training performance. The darker the color, the more statistically significant the correlation. "+" or "-" means positive or negative correlation. Overall, the total score (i.e., the sum of three game scores) has significant positive correlations with rotation, fitness, and overall score. This is likely a result of the strong contribution of Find Treasure since other tasks only show positive correlations with certain cognitive scores but not statistically significant.

The Find Treasure score has a significant positive correlation with theory of behavior, mental rotation, and fitness scores, suggesting that subjects who performed better on these tests also trained better agents. This is likely due to the need for good spatial reasoning and future behavior estimation in Find Treasure. The 1v1 Hide-and-Seek score shows a significant positive correlation with reflex score, likely due to the rapid changes in environment and task dynamics in the complex adversarial setting. Subjects with faster reaction speeds provided timely feedback to align with relevant state-action pairs. Overall, the total score (i.e., the sum of three game scores) has significant positive correlations with rotation, fitness, and overall score, suggesting that these cognitive skills are most relevant to the performance of c-Deep TAMER-guided agents.

## 5 Conclusion, Limitation, and Future Work

We introduce CREW for facilitating Human-AI teaming research from diverse human and machine learning scientific communities. CREW offers extensible environment design, enables real-time human-AI communication, supports hybrid Human-AI teaming, parallel sessions, multimodal feedback, and physiological data collection, and features ML community-friendly algorithm design. We also provide a set of built-in tasks and baseline algorithm implementations. Using CREW, we benchmarked a state-of-the-art human-guided RL algorithm against baseline methods involving 50 human subjects and provided insights into the relationship between individual human differences and agent-guiding performance.

CREW is still in the early efforts among several critical aspects. Future work will explore building more diverse and challenging tasks, including multiplayer tasks with complex strategies and robotics environments requiring an understanding of physics. While we have only benchmarked several algorithms, we hope CREW can help benchmark many existing algorithms that were not fully open-sourced in a unified environment. Finally, more supports on human physiological data processing and analysis shall be investigated and supported in CREW.

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

# 6 Appendix

## 6.1 Accessing CREW

Our fully open-sourced code base and detailed documentations can be found at `https://github.com/generalroboticslab/CREW.git`.

## 6.2 Platform details

The environments of CREW is implemented using `Unity 2021.3.24f1`, with packages `ML Agents 2.3.0-exp.3` Juliani et al. (2018), `Netcode for GameObjects 1.3.` net and `Nakama Unity 3.6.0.` nak. Algorithms are developed with `torchrl 0.3.0` Bou et al. (2023).

## 6.3 Hyperparameters

The hyperparameter settings for our experiments is summarized in Table. 4.

Table. 4: Hyperparameters

|  | c-Deep TAMER | DDPG | SAC |
|---|---|---|---|
| $\gamma$ | 0.99 | 0.99 | 0.99 |
| learning rate | 1e-4 | 1e-4 | 1e-4 |
| max_grad_norm | 0.1 | 0.1 | 0.1 |
| batch size | 16 | 240 | 240 |
| frames per batch | 8 | 240 | 240 |
| alpha_init | - | - | 0.1 |
| target entropy | - | - | -6.0 |
| actor scale_lb | - | - | 1e-4 |
| # Q value nets | - | 2 | 2 |
| target update polyak | 0.995 | 0.995 | 0.995 |
| actor exploration noise | $\mathcal{N}(0, 0.1)$ | $\mathcal{N}(0, 0.1)$ | - |
| credit assignment window | bowling[0.2, 4], others[0.2, 1] | - | - |

## 6.4 Algorithm Statistics

In Table. 5 we report the average number of updates of each algorithm. In the task that c-DeepTAMER showed the most advantage, Find Treasure, c-Deep TAMER had a relatively lower number of average updates than the RL baselines.

Table. 5: Average number of updates

|  | Bowling | Find Treasure | Hide and Seek |
|---|---|---|---|
| c-Deep TAMER | 1357.54 | 1114.72 | 1224.0 |
| DDPG | 600 | 1200 | 1200 |
| SAC | 600 | 1200 | 1200 |

## 6.5 Additional Individual Differences Results

## 6.6 Full Cognitive Test Analysis

We include the full results for the cognitive test and c-Deep TAMER score analysis. All linear regression plots including coefficients and p-values and summarized in Figure. 12.

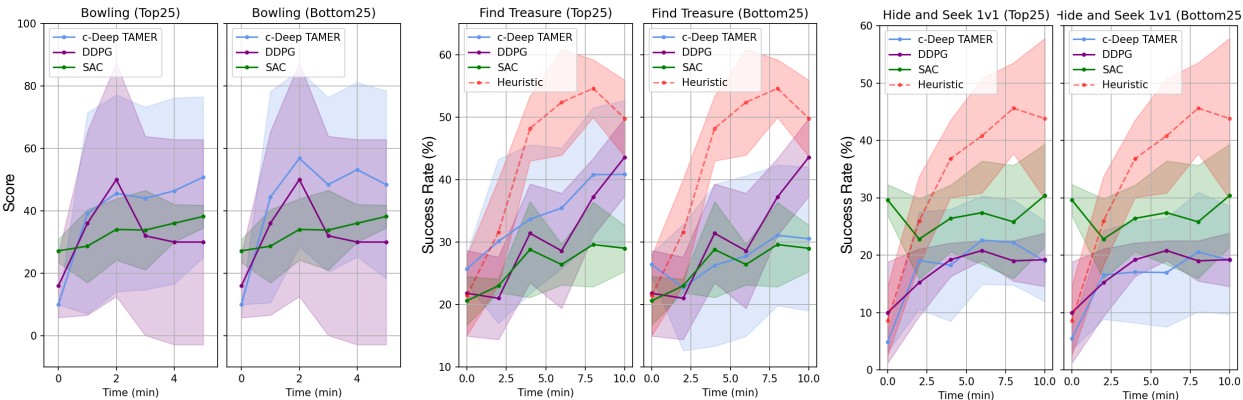

Figure. 11: Guided AI performance of the top and bottom 25 cognitive test scorers.

## 6.7 Instruction Video Script

**General Introduction:**

Welcome to the XXX Human AI Collaboration Experiment. Today, our session will start with preliminary cognitive and gaming proficiency tests to gauge your initial skills. Following this, we will delve into the main experiments where you will interact with AI algorithms through a series of engaging games. Our session will conclude with a short survey to capture your feedback on the experience. Your participation is invaluable in advancing our understanding of human AI interactions. Thank you for joining us today, and let's begin.

**Cognitive Test Introduction:**

Welcome to the Cognitive Test segment of our experiment. In this session, you will participate in five interactive games designed to assess various cognitive skills for about seven minutes. These tests will challenge your precision, reflexes, predictive abilities, problem-solving skills, and spatial awareness through engaging activities. Each test is brief, and you'll receive clear instructions before each one begins. Between each trial of each game, there will be a three-second intertrial interval where the computer screen will turn white with a Gray cross in the middle. Please focus on the center of the cross as much as possible during this time. Let's get started and see how you do.

**Eye Alignment Instruction:**

In this experiment, your goal is to align the ball positioned on the left side of the screen with the square on the right as accurately as possible. Each trial lasts for five seconds, and you'll have a total of six trials. Use your mouse to drag the ball during each trial. The time bar at the top center of the screen shows how much time you have for each trail. As time goes down, the bar gets smaller and changes color from green to red.

**Reflex Instruction:**

In this experiment, after a three-second countdown, the screen turns yellow. You must click as quickly as possible when the screen changes from yellow to green. Clicking during the yellow phase will result in a failure, and failing to click within two seconds after the screen turns green will also fail. You'll have a total of 6 trials.

**Theory of Behavior Instruction:**

In this experiment, watch the red ball moving for five seconds, then guess where it will be one second after it pauses. Click on the map to mark your prediction. You have only one chance to click. The closer you are, the higher your score. Remember, you only have two seconds to make your guess after the ball pauses. There will be 6 trials in total. The time bar at the top left shows the time for each trail's observation and prediction as time decreases. The bar shrinks and changes from green to red.

**Mention Rotation and Mental Fitting Instruction:**

In this experiment, you will need to answer 12 questions in total, each with only one correct answer. Click the button to choose the option that you think best answers the question. For each question, you have 8 seconds to view and respond. The time bar at the top right of the screen indicates how much time you have for each question. As time decreases, the bar shrinks and changes color from green to red.

**Spatial Mapping Instruction:**

In this experiment, you will need to answer six questions in total, each with only one correct answer. Watch the video and click the button to choose the option that you think best answers the question. For each question, you are free to answer while the video is playing, and you will also have three extra seconds after the video is paused to answer the question. The time bar at the top right of the screen indicates how much time you have for each question. As time decreases, the bar shrinks and changes color from green to red.

**Human Guiding AI Introduction:**

Welcome to the Human-Guiding AI section of the experiment. You will guide the AI to play three games for about 30 minutes in total in this section. You'll give feedback to the AI agents based on their performance in each game. Before starting each game, please enter your name in the name box and click on the Connect button. A match list will appear. Click on the Match button in the list. It will turn brown when clicked. Then click the Join Match button to enter the match. Once you've joined the match, select the AI agent in the Active Clients panel at the top right corner of the screen by clicking on it. The agent button will turn brown to indicate that you've selected it successfully. Now you can observe the agents and provide feedback. You'll constantly click on the positive and negative buttons to inform the agent of its performance. Positive means it's doing well. Negative means it's doing poorly.

**Bowling Instruction:**

Each episode of this bowling game consists of 10 rolls. At the start of each roll, there are 10 pins positioned on the right side of the screen. The agent then launches a ball in an attempt to knock down the pins, with the number of pins knocked down determining the score for that roll. The total score for the episode is calculated as the sum of the scores from all 10 rolls. Please guide the AI agent to maximize the total score across all rolls.

**Find Treasure Instruction:**

In this game, the AI agent, represented by the green character, begins with a limited field of view, only able to perceive surroundings, while the rest of the map is obscured by shadow. As the agent navigates the map, areas it has visited become revealed within its field of view. Please guide the AI agent to locate the treasure and navigate to it, represented by the brown chest, as quickly as possible.

**1v1 Hide-and-Seek Instruction:**

In this game, the AI agent, represented by the red character, begins with a limited field of view, only able to perceive surroundings, while the rest of the map is obscured by

shadow.  As the agent navigates the map, areas it has visited become revealed within its field of view.  Guide the AI agent to chase the hider, represented by the green character, and catch it as quickly as possible.

### 6.7.1  Compensation

We pay each human subject $20 for participation.

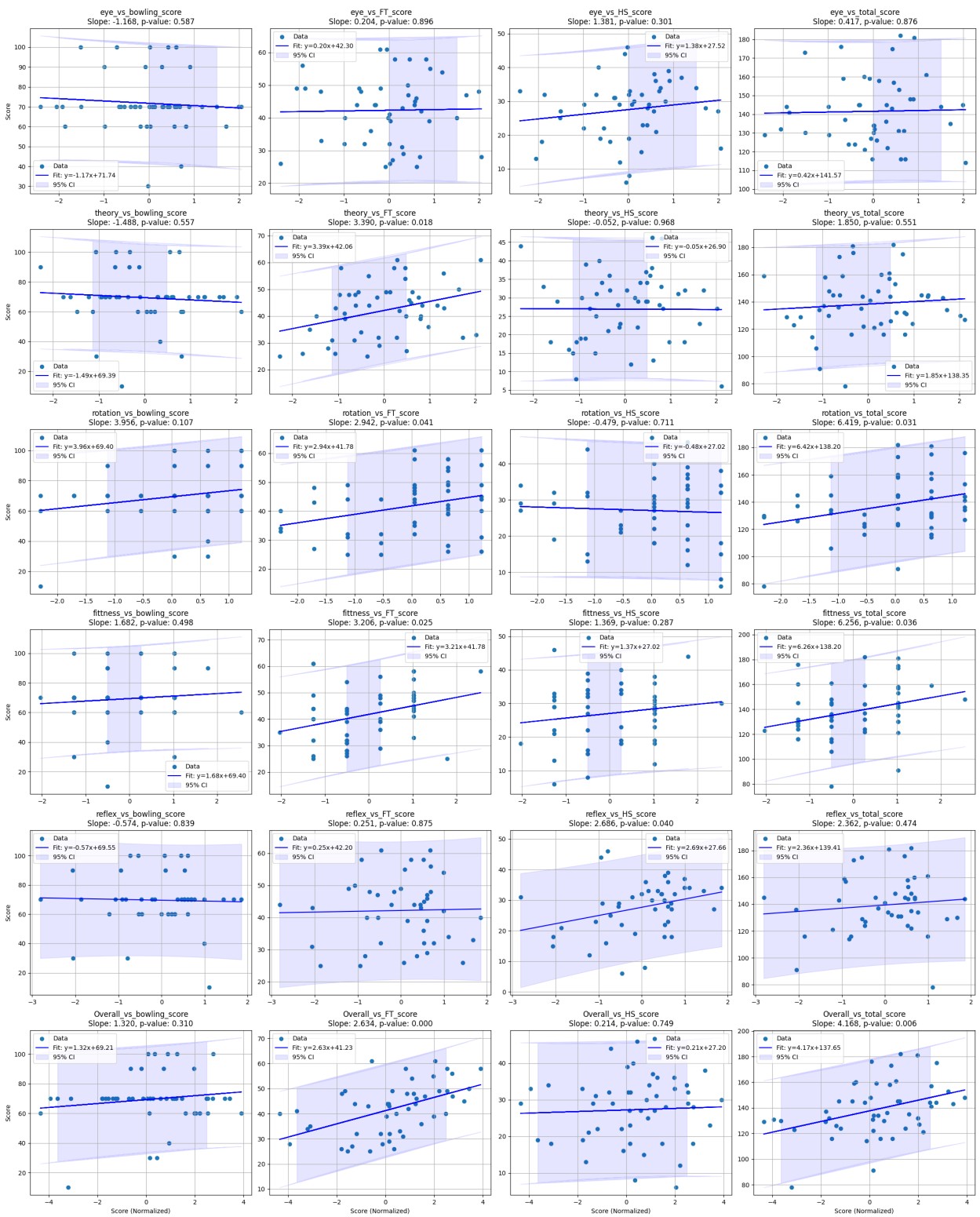

Figure. 12: Linear Regression Plots of Correlation Analysis

