# OpenReview forum: "CREW: Facilitating Human-AI Teaming Research"
_TMLR — Accepted by TMLR_

### Review · Reviewer_5saz · 2024-10-21

**Summary Of Contributions:**

This paper introduces CREW, a platform to facilitate Human-AI teaming research. The authors design a novel platform for human-AI collaboration which tackles some of the limitations of the existing platforms. Furthermore, the paper contains a few developed environments and a benchmarking study.

**Audience:**

Yes

**Claims And Evidence:**

No

**Requested Changes:**

1. The introductory section places significant emphasis on the teaming aspect, yet the rest of the paper seems more focused on human-guided learning for AI algorithms. I would like to ask the authors for clarification on this choice. While it's clear to me how human data and feedback can be beneficial in the AI space, I find it less clear how AI can contribute to human studies. Specifically, what value does this platform offer to neuroscientists or other researchers in human-centered fields?

2. The paper provides almost no information about the hardware requirements necessary for successfully using the platform. Figure 7 depicts a setup for human data collection, but I believe additional details about the hardware used to gather these data would help other labs adopt the platform more easily. I also have a few questions regarding the computational resources mentioned in your experiments. The statement, "All evaluations were run on a headless server with 8 × NVIDIA RTX A6000," is somewhat confusing. What exactly do you mean by "evaluation"? What resources were required to train the reinforcement learning algorithms? Given that your networks do not appear large enough to justify the use of 8 GPUs, this point needs further clarification.

3. I quickly went through your anonymized code base. I believe that in these types of works, the code is very important. The code is not well documented and does not seem ready for publication.

**Strengths And Weaknesses:**

# Strengths
The work is both interesting and valuable to the community. The limitations of previously developed platforms are clearly articulated, and the authors provide a thorough yet concise description of the strengths of their novel platform in relation to the existing literature.

Several of the proposed features are particularly noteworthy for the machine learning community, especially in the area of human-assisted algorithms.

Overall, the paper is well-written and easy to follow.


# Weaknesses
The work defines CREW as a Human-AI teaming research platform. Given this definition, I would have expected a platform that facilitates true collaboration between human and AI agents, where each plays distinct roles to tackle complex tasks that would otherwise be extremely difficult to address.

Unfortunately, I don't believe this work fully lives up to that premise. Despite the authors' efforts to emphasize Hybrid Human-AI teaming, the tasks and study presented in the paper do not clearly demonstrate this collaborative aspect. Instead, the platform seems primarily geared toward scaling the collection of human behavioral and physiological data for human studies and improving the integration of human feedback and prior knowledge into machine learning algorithms.

---

> ### Author Response · Authors · 2024-10-21
> **Response to Reviewer 5saz**
>
> We thank the reviewer for your recognition of our unique contributions to machine learning and your thoughtful comments. We would like to address all of your concerns and questions below with point responses.
>
> (part 1)
>
> ----
> > “Unfortunately, I don't believe this work fully lives up to that premise. Despite the authors' efforts to emphasize Hybrid Human-AI teaming, the tasks and study presented in the paper do not clearly demonstrate this collaborative aspect. Instead, the platform seems primarily geared toward scaling the collection of human behavioral and physiological data for human studies and improving the integration of human feedback and prior knowledge into machine learning algorithms.”
> >“The introductory section places significant emphasis on the teaming aspect, yet the rest of the paper seems more focused on human-guided learning for AI algorithms. I would like to ask the authors for clarification on this choice. While it's clear to me how human data and feedback can be beneficial in the AI space, I find it less clear how AI can contribute to human studies. Specifically, what value does this platform offer to neuroscientists or other researchers in human-centered fields?”
>
> We acknowledge that this is a great observation. Indeed, most established and reproducible baselines only exist for human-guided RL, and many more studies need to be done in other aspects of Human-AI teaming. Some primary studies have initiated exploration on how the introduction of AI teammates will affect overall team performance [1-5] and the psychological and cognitive impact on humans [6-9], but concrete research outcomes still remain a large gap. This is exactly our motivation behind designing CREW. In discussions with many researchers in this field, including neuroscientists and other researchers in human-centered fields, a common constraint of running larger-scale and diverse experiments is the lack of such a platform considering both humans and AI agents. Hybrid, collaborative Human-AI teaming involves 1) building extensible tasks and environments,  2) running different learned policies, 3) hosting multiple AI and human players at the same time, and 4) collecting data from both human and AI agents. CREW addresses these essential needs to facilitate these broader aspects of Human-AI teaming: 1) CREW provides an environment infrastructure based on a user-friendly game engine and includes built-in example tasks; 2) CREW supports both state-of-the-art RL and human-guided RL algorithm examples, and infrastructures to implement more; 3)CREW offers possibilities to easily host multiple human agents and AI agents at the same time, as demonstrated by our example tasks; 4) CREW further provides Human-AI interaction interfaces and data collection mechanisms for both human data and AI agent data.
>
> When we constructed CREW’s human data collection modules, we consulted existing human-centered research in human-human teaming and Human-AI teaming. These include using human eye gaze or EEG data to enhance vision models and policies [10-12], human pupilometry and EEG data for brain synchrony[13], human-robot interaction with service robots [14], and physiological data-informed team performance prediction [15]. CREW supports the data collection of these existing works and provides a flexible interface to connect with more devices.
>
> Moreover, our experiments in Section 4.2 also consider human-centered aspects. Specifically, we studied  how individual human differences would affect the AI policies guided by them. This research question is inspired by neuroscience and cognitive science studies on individual differences. In fact, we are the first to conduct such large-scale human studies to provide insights about humans under human-guided RL contexts due to the capabilities of CREW.
>
> We chose human-guided RL as the task for demonstrating the usability of CREW due to its  accessible prior work. Human-guided RL is also among the more mature subfields of Human-AI teaming and has demonstrated strong benefits in challenging gameplays [16-19], robotics [20], self-driving [21], and many more. It’s also been popularized by the recent rise of RLHF in large language models. Despite these advancements, there have not been enough studies on reinforcement learning from real-time human feedback and not enough understanding of how humans affect and get affected by this process. With all the above motivations, we chose to use human-guided RL as the demonstration task. However, we would like to clarify thatthis certainly does not mean the CREW is limited only to this research purpose, as we illustrated above on the functionality, motivation, and example tasks and human-centered experiments
>
> Overall, our key contribution in CREW is to provide a novel, modularized, and flexible infrastructure with large-scale Human-AI studies to demonstrate its capability. The goal is to help facilitate richer research in Human-AI teaming, in both human-guided RL and beyond.

---

> > ### Author Response · Authors · 2024-10-21
> > **Response to Reviewer 5saz (part2)**
> >
> > (part2)
> >
> > We will clarify the above points in our revised paper.
> >
> >
> > > “The paper provides almost no information about the hardware requirements necessary for successfully using the platform. Figure 7 depicts a setup for human data collection, but I believe additional details about the hardware used to gather these data would help other labs adopt the platform more easily.”
> >
> > Hardware requirements are dependent on the specific Human-AI experiment that the user wishes to conduct. What we aim to provide in CREW is the infrastructure for quickly setting up Human-AI experiments, rather than specific experiment conditions, as it is impossible to foresee the needs of all potential research problems. The minimal essential requirement for single-modal experiments like Deep TAMER is simply a desktop with reasonable computation power and a mouse and keyboard.
> >
> > > “I also have a few questions regarding the computational resources mentioned in your experiments. The statement, "All evaluations were run on a headless server with 8 × NVIDIA RTX A6000," is somewhat confusing. What exactly do you mean by "evaluation"? What resources were required to train the reinforcement learning algorithms? Given that your networks do not appear large enough to justify the use of 8 GPUs, this point needs further clarification.”
> >
> > By “evaluation”, we are referring to rolling out the trained models on test environments with unseen initial configurations and multiple random seeds. Unlike many experiment setups in reinforcement learning where training and evaluation are performed in the same environment, we take intermediate checkpoints and roll out them on held-out test conditions, which would require more computational resources for multiple test conditions and random seeds. We would like to note that this is only for speeding up the evaluation to provide full results reported in our paper, and it’s a result of our large-scale human study conducted in our paper,  but running each policy does not require such resources, no matter in training or evaluation. As in Section 4.1, human-guided RL training is done on a desktop with a single NVIDIA RTX 4080 GPU.  We will clarify this detail in our revised paper.
> >
> > > “I quickly went through your anonymized code base. I believe that in these types of works, the code is very important. The code is not well documented and does not seem ready for publication.”
> >
> > Thank you for pointing this out. We did have fully prepared documentation to be released when the paper is published, but we have now also created a version for your reference that can be accessed at https://crewplatform.github.io/crew-docs/index.html. We have also updated the README of the repo to provide an overview of the code base.

---

> > > ### Author Response · Authors · 2024-10-21
> > > **Response to Reviewer 5saz (references)**
> > >
> > > ----
> > > *References*
> > >
> > > [1] Flathmann, Christopher, et al. "Examining the impact of varying levels of AI teammate influence on human-AI teams." International Journal of Human-Computer Studies 177 (2023): 103061.
> > >
> > > [2] Shaikh, Sonia Jawaid, and Ignacio F. Cruz. "AI in human teams: effects on technology use, members’ interactions, and creative performance under time scarcity." AI & SOCIETY 38.4 (2023): 1587-1600.
> > >
> > > [3] Dell'Acqua, Fabrizio, Bruce Kogut, and Patryk Perkowski. "Super mario meets ai: Experimental effects of automation and skills on team performance and coordination." Review of Economics and Statistics (2023): 1-47.
> > >
> > > [4] Seeber, Isabella, et al. "Machines as teammates: A research agenda on AI in team collaboration." Information & management 57.2 (2020): 103174.
> > >
> > > [5] Zhang, Rui, et al. "" An ideal human" expectations of AI teammates in human-AI teaming." Proceedings of the ACM on Human-Computer Interaction 4.CSCW3 (2021): 1-25.
> > >
> > > [6] Mallick, Rohit, et al. "The pursuit of happiness: the power and influence of AI teammate emotion in human-AI teamwork." Behaviour & Information Technology (2023): 1-25
> > >
> > > [7] Musick, Geoff, et al. "What happens when humans believe their teammate is an AI? An investigation into humans teaming with autonomy." Computers in Human Behavior 122 (2021): 106852.
> > >
> > > [8] Zhang, Guanglu, et al. "Trust in an AI versus a Human teammate: The effects of teammate identity and performance on Human-AI cooperation." Computers in Human Behavior 139 (2023): 107536.
> > >
> > > [9] Iannone A, Giansanti D. Breaking Barriers-The Intersection of AI and Assistive Technology in Autism Care: A Narrative Review. J Pers Med. 2023 Dec 28;14(1):41. doi: 10.3390/jpm14010041. PMID: 38248742; PMCID: PMC10817661.
> > >
> > > [10] Ravi Kumar Thakur, MD Sunbeam, Vinicius G Goecks, Ellen Novoseller, Ritwik Bera, Vernon Lawhern, Greg Gremillion, John Valasek, and Nicholas R Waytowich. Imitation learning with human eye gaze via multi-objective prediction.
> > >
> > > [11] Duo Xu, Mohit Agarwal, Ekansh Gupta, Faramarz Fekri, and Raghupathy Sivakumar. Accelerating reinforcement learning using eeg-based implicit human feedback. Neurocomputing, 460:139–153, 2021.
> > >
> > > [12] Sharath Koorathota, Nikolas Papadopoulos, Jia Li Ma, Shruti Kumar, Xiaoxiao Sun, Arunesh Mittal, Patrick Adelman, and Paul Sajda. Fixating on attention: Integrating human eye tracking into vision transformers. arXiv preprint arXiv:2308.13969, 2023
> > >
> > > [13] Suzanne Dikker, Lu Wan, Ido Davidesco, Lisa Kaggen, Matthias Oostrik, James McClintock, Jess Rowland, Georgios Michalareas, Jay J Van Bavel, Mingzhou Ding, et al. Brain-to-brain synchrony tracks real-world dynamic group interactions in the classroom. Current biology, 27(9):1375–1380, 2017.
> > >
> > > [14] Judith S Heinisch, Jérôme Kirchhoff, Philip Busch, Janine Wendt, Oskar von Stryk, and Klaus David. Physiological data for affective computing in hri with anthropomorphic service robots: the affect-hri data set. Scientific Data, 11(1):333, 2024.
> > >
> > > [15] Yinuo Qin, Weijia Zhang, Richard Lee, Xiaoxiao Sun, and Paul Sajda. Predictive power of pupil dynamics in a team based virtual reality task. In 2022 IEEE Conference on Virtual Reality and 3D User Interfaces Abstracts and Workshops (VRW), pp. 592–593. IEEE, 2022.
> > >
> > > [16] Knox, W. Bradley, and Peter Stone. "Interactively shaping agents via human reinforcement: The TAMER framework." Proceedings of the fifth international conference on Knowledge capture. 2009.
> > >
> > > [17] Warnell, Garrett, et al. "Deep tamer: Interactive agent shaping in high-dimensional state spaces." Proceedings of the AAAI conference on artificial intelligence. Vol. 32. No. 1. 2018.
> > >
> > > [18] Arakawa, Riku, et al. "Dqn-tamer: Human-in-the-loop reinforcement learning with intractable feedback." arXiv preprint arXiv:1810.11748 (2018).
> > >
> > > [19] Arumugam, Dilip, et al. "Deep reinforcement learning from policy-dependent human feedback." arXiv preprint arXiv:1902.04257 (2019).
> > >
> > > [20] Wang, Xiaofei, et al. "Skill preferences: Learning to extract and execute robotic skills from human feedback." Conference on Robot Learning. PMLR, 2022.
> > >
> > > [21] Kim, Jinkyu, et al. "Grounding human-to-vehicle advice for self-driving vehicles." Proceedings of the IEEE/CVF conference on computer vision and pattern recognition. 2019.

---

> > > > ### Author Response · Authors · 2024-11-06
> > > > **We would love to hear from you if you have further questions**
> > > >
> > > > Dear reviewer, thank you again for your thoughtful review for our manuscript. We are following up to see if you have further questions about our paper. We aim to try our best to address your concerns of our paper. Thank you again!

---

### Review · Reviewer_F7Uw · 2024-10-23

**Summary Of Contributions:**

The paper introduces CREW, a platform designed to facilitate Human-AI teaming research. The platform aims to address several challenges, including the limited number of tasks, real-time human feedback during training, hybrid Human-AI teaming support, parallel sessions for scalable experiments, and comprehensive human and agent data collection. The authors demonstrate the platform's capabilities through a benchmarking study involving 50 human subjects.

**Audience:**

Yes

**Claims And Evidence:**

Yes

**Requested Changes:**

Critical requested changes:
- please highlight somewhere the number of network/policy updates for each algorithm, if those are different, you should also report the results of the baselines with equals number of updates
- the "heuristic" baseline should be better explained as well as why it is expected to has the highest performance, I understood it as an online DDPG with offline human feedbacks/reward function (unlike "c-Deep TAMER" that would receive the feedback incrementally)
- the "heuristic" baseline is missing on Bowling
- except if I misunderstood, the "heuristic" baseline should not be the same across "top 15" and "all 50", I assume what is displayed now is "all 50"

Non critical ones:
- it would be nice to clearly state that each trained RL agent is only receiving the feedback of a single human
- what are the hyperparameters for the heuristic DDPG?
- please mention how long the humans have to wait for the update step?
- "this positive correlation between cognitive test scores and guided AI performance holds true on all tasks, as shown in Figure. 9" I failed to see that on Bowling (except for the checkpoint at 1 min. all the others look like the opposite)
- it would also be interesting to know the upper bound performance of SAC and DDPG if we trained them longer

**Strengths And Weaknesses:**

Strengths:
- Comprehensive Design: CREW is designed to be extensible and modular. It is also open-source and cross-platform.
- Real-Time Interaction: The platform supports real-time communication between humans and AI agents, with real time feedback during training.
- Data Collection: CREW collects a variety of data, including physiological signals.
- Human Study: The authors conducted the largest study with 50 persons for real-time training, demonstrating the platform's practical usability.

Weaknesses:
- it is not clear if the benefits of c-Deep TAMER come from the higher number of network updates or the actual human feedback (the batch size is 15 times smaller than DDPG/SAC, which let me think the number of network updates is also 15 times bigger for c-Deep TAMER)
- the experiment section should show the importance of the framework, I believe the real-time training is a key contribution but those experiments only show that offline training is more beneficial ("heuristic" vs "c-Deep TAMER" in Figure 9.)
- the authors claims that other works support only a single task, but the 4 proposed tasks in CREW are also a bit limited and do not really show the need for real-time human feedback
- CREW seems to offer visual and low level observations but nothing for textual information
- 5 to 10 minutes of training seems to be very small for RL training

---

> ### Author Response · Authors · 2024-10-23
> **Response to Reviewer F7Uw**
>
> We thank the reviewer for the thoughtful comments. We would like to address all of your concerns and questions below with point responses.
>
> (part 1)
>
> ----
> > “it is not clear if the benefits of c-Deep TAMER come from the higher number of network updates or the actual human feedback (the batch size is 15 times smaller than DDPG/SAC, which let me think the number of network updates is also 15 times bigger for c-Deep TAMER)”
>
> We attach the average number of updates of each algorithm here for reference. In the task that c-DeepTAMER showed the most advantage, Find Treasure, c-Deep TAMER had a relevantly lower number of average updates than the RL baselines. We agree that in some other cases, this could be a factor affecting c-Deep TAMER’s performance. However, we would like to clarify that c-Deep TAMER is not a novel algorithm that we propose, but rather an implementation of existing state-of-the-art that many prior works have used consistently as a strong baseline, and we benchmark its performance to test the capability of CREW by upgrading some of its practical implementations to catch up with the latest DL/RL practices. Specifically, we implemented c-Deep TAMER by closely following the original Deep TAMER paper with only extensions to a continuous action space and reported its performance for benchmark purposes.
>
> | Algorithm             | Bowling | Find Treasure | Hide and Seek |
> | :---------------- | :------: | :------: | :------: |
> | c-Deep TAMER| 1357.54  | 1114.72 | 1224.0 |
> | DDPG | 600 | 1200 | 1200 |
> | SAC | 600 | 1200 | 1200 |
>
> > “the experiment section should show the importance of the framework, I believe the real-time training is a key contribution but those experiments only show that offline training is more beneficial ("heuristic" vs "c-Deep TAMER" in Figure 9.)”
>
> Yes, real-time training is one of the key features that we enable in CREW. To clarify, the “heuristic” method is a carefully hand-crafted dense reward by human experts and does not exist in real-world problems. We meant to use the “heuristic” method to show the upper-bound limit if an expert exists and ample time is allowed. In other words, the closer other methods get to it, the better these methods should be The main purpose of real-time training is to avoid reward engineering, which requires domain expertise and extensive trial-and-error, and make efficient RL training more accessible to a broader user group.   Moreover, our human-guided RL (c-DeepTAMER) experiments were conducted without any requirement or assumptions about human expertise, which is much more accessible than the “heuristic” upper-bound method.
>
> >”the authors claims that other works support only a single task, but the 4 proposed tasks in CREW are also a bit limited and do not really show the need for real-time human feedback”
>
> Our key contribution in CREW is building the infrastructure to allow researchers to quickly develop Human-AI teaming experiments at scale. One of the key capabilities is to allow rapid prototyping with extensible environment support. We have demonstrated this in our examples in Figure 2, 3, and 4.  In the experiments where we aimed to demonstrate the capability of CREW, we are bound by the current status of research progress in Human-AI teaming. Therefore, we chose the most developed and popular paradigm, human-guided RL, as the example task. As we discussed in the paper, the tasks we used in our algorithms are already more complex than existing practices, which is further evidenced by the performance gap in our results. We would like to clarify that the countless applications of Human-AI teaming make it impossible to foresee all potential research problems. Therefore, instead of attempting to build a wide range of environments, especially much beyond what the current algorithms can achieve, we provide the infrastructure that helps developers equip their tasks with Human-AI teaming features, including extensible environments, real-time multi-human-agent interaction, and data collection. The 4 proposed tasks are example environments where users can build on top. Compared with previous works where the task is limited to one with closed-source implementation, we believe that CREW offers strong promises for scalable efforts.

---

> > ### Author Response · Authors · 2024-10-23
> > **Response to Reviewer F7Uw (part 2)**
> >
> > (part 2)
> > >”CREW seems to offer visual and low level observations but nothing for textual information”
> >
> > In our demonstrated tasks, we offer only visual and low-level state observations. However, the infrastructure can easily support textual information such as text inputs or connections to LLMs. For example, the data recording and synchronization interface is device agnostics that can support audio input. We made the decision to demonstrate visual and low-level observations since these are the most common observation spaces in the reinforcement learning domain. We will add textual information in our future work in our revised paper.
> >
> > >”5 to 10 minutes of training seems to be very small for RL training”
> >
> > Yes, as shown in our results in Figure 9, as the task difficulty increases, the gap between RL and the “heuristic” upper bound of human feedback training increases. Comparatively, however, human-guided RL shows stronger efficiency in boosting the training. We would like to clarify that the relative performance within the same time is what we aim to emphasize. The total time is the same for all algorithms. We decided to use this time due to the large-scale human studies across multiple tasks as well as the cognitive studies (1 hour per subject).
> >
> > >”please highlight somewhere the number of network/policy updates for each algorithm, if those are different, you should also report the results of the baselines with equals number of updates”
> >
> > The average number of updates for each algorithm is listed in the table below. We will include this in the revised paper. As shown here, the number of updates among these algorithms almost matches for the more complex tasks in Find Treasure and Hide&Seek.
> > | Algorithm             | Bowling | Find Treasure | Hide and Seek |
> > | :---------------- | :------: | :------: | :------: |
> > | c-Deep TAMER| 1357.54  | 1114.72 | 1224.0 |
> > | DDPG | 600 | 1200 | 1200 |
> > | SAC | 600 | 1200 | 1200 |
> >
> > >”the "heuristic" baseline should be better explained as well as why it is expected to has the highest performance, I understood it as an online DDPG with offline human feedbacks/reward function (unlike "c-Deep TAMER" that would receive the feedback incrementally)”
> >
> > We would like to clarify that the “heuristic” baseline does not come with any human feedback. It’s a standard RL method with extensively tuned dense reward functions through trial and error and expert knowledge of the tasks. Specifically, in our experiments, we tuned the  “heuristic” reward function through a combination of exploration reward and distance reward. I
> > We would like to highlight that these rewards do not naturally exist in real-world problems, as they require domain expertise and extensive trial-and-error. We show the “heuristic” method as an upper bound, indicating what perfect human feedback can achieve. In contrast, the human-guided RL algorithms only rely on online human feedback from non-experts and the sparse reward function without reward tuning.
> >
> > >the "heuristic" baseline is missing on Bowling
> >
> > Due to the difficulty of designing a vision-based dense feedback for Bowling, we did not implement this baseline, which exactly reflects our motivation for human-AI teaming Though the task of Bowling does not appear simple, a design reward function based on visual input does not exist and remains challenging to clearly define. On the other hand, human carries implicit intuitions on the performance of the task. Human-guided RL is the paradigm that can effectively leverage this, hence our implementation of a classical baseline, c-DeepTAMER, to benchmark it for future community usage in CREW.
> >
> > >”except if I misunderstood, the "heuristic" baseline should not be the same across "top 15" and "all 50", I assume what is displayed now is "all 50"”
> >
> > We would like to clarify that the dense feedback in the “heuristic” baseline is designed by domain experts and the reward function remains the same across experiments. Hence, it is unrelated to the human participants. We trained the heuristic baseline on 5 random seeds and reported the average results, consistent with the RL baselines.
> >
> > > “it would be nice to clearly state that each trained RL agent is only receiving the feedback of a single human”
> >
> > Thank you for pointing this out. We will make this clear in our revised paper.
> >
> > > “what are the hyperparameters for the heuristic DDPG?”
> >
> > The training hyperparameters for heuristic DDPG are the same as the DDPG baseline. The only difference is the additional dense feedback. We will clarify this in our revised paper.
> >
> > >”please mention how long the humans have to wait for the update step?”
> >
> > Since we focus on real-time human feedback, the human trainers do not have to wait for an update step. Instead, an update step happens whenever they choose to provide a feedback signal, as implemented in the original DeepTAMER paper.

---

> > > ### Author Response · Authors · 2024-10-23
> > > **Response to Reviewer F7Uw (part 3)**
> > >
> > > (part 3)
> > > >”"this positive correlation between cognitive test scores and guided AI performance holds true on all tasks, as shown in Figure. 9" I failed to see that on Bowling (except for the checkpoint at 1 min. all the others look like the opposite)”
> > >
> > > Thanks for pointing this out. This is a typo. We meant “Figure 10” instead of “Figure 9” As shown in the right column in Figure. 10, our linear analysis suggests that cognitive test scores positively correlate with overall guided AI performance on all tasks. We will fix this typo.
> > >
> > > >”it would also be interesting to know the upper-bound performance of SAC and DDPG if we trained them longer”
> > >
> > > Based on our experience, after a few hours, the RL baselines will eventually converge to a near 100 percent success rate. However, this process takes much longer, while the human-guided baseline can accelerate the training speed. On the other hand, our paper does not focus on proposing a strong human-guided RL algorithm. We instead implement the state-of-the-art baseline to demonstrate the capability of CREW and release it for future community usage to develop stronger algorithms.

---

> > > > ### Author Response · Authors · 2024-11-06
> > > > **We would love to hear from you if you have further questions**
> > > >
> > > > Dear reviewer, thank you again for your thoughtful review for our manuscript. We are following up to see if you have further questions about our paper. We aim to try our best to address your concerns of our paper. Thank you again!

---

> > > > > ### Comment · Reviewer_F7Uw · 2024-11-21
> > > > >
> > > > > Thank you for your answers. I indeed misunderstood the heuristic baseline. I do not have further questions.

---

> > > > > > ### Author Response · Authors · 2024-11-21
> > > > > > **Thank you for your response**
> > > > > >
> > > > > > We are glad to have clarified this. We will make this more clear in the revised paper. Thank you!

---

### Review · Reviewer_o49T · 2024-10-24

**Summary Of Contributions:**

This paper introduces a platform for research where humans and AIs collaborate to solve tasks in various game environments, which is useful for research into human-AI teaming. This platform has a variety of features, e.g. allowing multiplayer environments and human data collection. The paper also investigates whether humans who are better at cognitive tasks train AIs faster, and finds that they do. It also introduces a novel RL algorithm for training AIs with humans, that seems to be better than baselines in some settings.

**Audience:**

Yes

**Claims And Evidence:**

No

**Requested Changes:**

Please either remove the claim that "Subjects scoring higher on cognitive tests are generally better at guiding RL" or justify that that is true in more than c-Deep TAMER on Find Treasure. This is necessary for my recommendation for acceptance,

Please consider using more consistent statistical methods for the empirical results. This is not necessary but I recommend it.

Please consider rewriting the abstract to be clearer to more generalist ML researchers. This is not necessary.

**Strengths And Weaknesses:**

For context, I had not previously read about human-AI teaming, though I'm of course familiar with the general concept. So some of my comments here might be a result of that ignorance.

To me the most interesting results were the results on training performance and individual differences. However, the statistics used in this section seemed somewhat arbitrary to me: maybe this is just a result of me not knowing the research field very well, but is it normal to remove datapoints that are more than 1.5x the IQR away from the mean (in section 4.2)? Also, it seems arbitrary to compare the top 15 to all trainers; why not compare top 25 to bottom 25? This seems especially dubious because of the fact that you use correlation coefficients as the metric in Figure 10; why switch statistical method like that?

I am very confused about the evidence for the claim that "Subjects scoring higher on cognitive tests are generally better at guiding RL": in Figure 9, I don't see that effect except for c-Deep TAMER on Find Treasure. Figure 10 seems consistent with my interpretation here: overall cognitive ability only correlated significantly with agent performance on Find Treasure.

I found the submission somewhat confusing to understand based on the abstract. For example, I didn't understand what kind of tasks the human-AI teams would be doing (game tasks? programming tasks? NLP tasks?). The abstract also doesn't mention the empirical results.

---

> ### Author Response · Authors · 2024-10-25
> **Response to Reviewer o49T**
>
> We thank the reviewer for the thoughtful comments. We would like to address all of your concerns and questions below with point responses.
>
> (part 1)
> >”To me the most interesting results were the results on training performance and individual differences. However, the statistics used in this section seemed somewhat arbitrary to me: maybe this is just a result of me not knowing the research field very well, but is it normal to remove datapoints that are more than 1.5x the IQR away from the mean (in section 4.2)? Also, it seems arbitrary to compare the top 15 to all trainers; why not compare top 25 to bottom 25? This seems especially dubious because of the fact that you use correlation coefficients as the metric in Figure 10; why switch statistical method like that?”
>
> We would like to clarify our choices for the quantitative analysis. We follow Tukey’s method [1], a common statistical method to remove outliers beyond the 1.5 IQR. We chose the top 15 due to its proximity to the highest rankings while maintaining a reasonable sample size. We uploaded the top 25 vs the bottom 25 in our repo (https://github.com/crewplatform/CREW/blob/main/assets/top25_bottom25.pdf) for your reference. As indicated by the plot, our conclusions remain the same.
>
> We would like to clarify that both Figure 9 and Figure 10 arrive at the same conclusion with different emphasis. The difference is that Figure 9 shows the learning performance comparison with respect to the total cognitive scores along the entire training process, while Figure 10 shows a summarized learning performance but emphasizes the breakdown of different cognitive test tasks. Our choices of analysis techniques are based on different emphases in these two figures.
>
> >”I am very confused about the evidence for the claim that "Subjects scoring higher on cognitive tests are generally better at guiding RL": in Figure 9, I don't see that effect except for c-Deep TAMER on Find Treasure. Figure 10 seems consistent with my interpretation here: overall cognitive ability only correlated significantly with agent performance on Find Treasure.”
>
> Thank you for pointing this out. This is a typo. We meant “Figure 10” instead of “Figure 9”. We will fix this typo. As shown in the right column in Figure. 10, our linear analysis suggests that cognitive test scores positively correlate with overall guided AI performance on all tasks. We do agree that this correlation is only significant in Find Treasure. We will make this clear in our revised paper. We would also like to clarify that our paper does not focus on presenting a novel algorithm. Instead, c-DeepTAMER is a state-of-the-art baseline implementation. Our paper focuses on the contribution of the novel Human-AI teaming platform: CREW. These experiments and observations are meant to provide experimental mechanisms, infrastructure, and initial observations as potential considerations of experiments and benchmarks for future algorithm development.
>
> >”I found the submission somewhat confusing to understand based on the abstract. For example, I didn't understand what kind of tasks the human-AI teams would be doing (game tasks? programming tasks? NLP tasks?). The abstract also doesn't mention the empirical results.”
>
> We would like to clarify that we focus on real-time decision-making tasks. We will make this more clear in our revised paper. We did not mention empirical results because our main contribution is the platform (real-time human guidance, hosting multiple humans and AI agents, extensive environments, networking, human data and AI data collection, etc.) and the ability to benchmark previous algorithms to facilitate future development in this emerging area. The specific performance of the algorithms is not a focus of discussion in this work. The algorithms are implementations of existing state-of-the-art approaches.
>
> >”Please either remove the claim that "Subjects scoring higher on cognitive tests are generally better at guiding RL" or justify that that is true in more than c-Deep TAMER on Find Treasure. This is necessary for my recommendation for acceptance,”
>
> Thank you for pointing this out. We will remove this claim and reword this as “Cognitive test scores are positively correlated with c-Deep TAMER guidance performance in Find Treasure”. We will also clarify that this experiment is meant to provide a mechanism and a novel setup for future algorithm development to consider, instead of proving a general conclusion for all human-guided RL algorithms. We believe that this setup can help investigate differences between AI algorithm design and human individual differences. We will clarify these points in the revised paper.

---

> > ### Author Response · Authors · 2024-10-25
> > **Response to Reviewer o49T (part 2)**
> >
> > (part 2)
> > >”Please consider using more consistent statistical methods for the empirical results. This is not necessary but I recommend it.”
> >
> > As discussed above, we presented an analysis using multiple methods due to the different emphasis of analysis, which suggests the same conclusion. We followed standard statistical methods.
> > >”Please consider rewriting the abstract to be clearer to more generalist ML researchers. This is not necessary.”
> >
> > Thank you for this suggestion. We will be more clear in our wording in our revised version as we discussed above
> >
> > ----
> > *Reference*
> >
> > [1] Tukey, John W. "Exploratory data analysis." Reading/Addison-Wesley (1977).

---

> > > ### Comment · Reviewer_o49T · 2024-10-30
> > > **Response to authors**
> > >
> > > Thanks for your response.
> > >
> > > I still think your claims about the human experiment results are too strong.
> > >
> > > You write:
> > >
> > > > Overall, the total score (i.e., the sum of three game scores) has significant positive correlations with rotation, fitness, and overall score
> > >
> > > This is true, but isn't it only true because of the Find Treasure result? If so, I'd prefer you'd be clear that this result is only true because of Find Treasure (and it's not a significant effect in either of the other environments).
> > >
> > > I have a similar concern with
> > >
> > > >  The overall correlation between cognitive test scores and guided AI performance is statistically significant (∗∗, p < 0.01).
> > >
> > > Is that true if you ignore Find Treasure?
> > >
> > >
> > > Thanks for that top25/bottom25 graph. I like that graph, I think it would make the paper stronger if you used that instead of the current Figure 9.

---

> > > > ### Author Response · Authors · 2024-10-30
> > > > **Response to Reviewer o49T**
> > > >
> > > > Thank you for your response. We will address each of your points below:
> > > >
> > > > >"This is true, but isn’t it only true because of the Find Treasure result? If so, I’d prefer you’d be clear that this result is only true because of Find Treasure (and it’s not a significant effect in either of the other environments)."
> > > >
> > > > Thank you for your suggestion. We will modify this sentence into:
> > > > "Overall, the total score (i.e., the sum of three game scores) has significant positive correlations with rotation, fitness, and overall score. This is likely a result of the strong contribution of Find Treasure since other tasks only show positive correlations with certain cognitive scores but not statistically significant." We will also add a clarification statement in our revised paper to state that we are not making a general conclusion yet, and the goal of our experiments is to help provide a useful metric and experiment to study individual differences for future work. We report our findings as an example instead of offering a fixed conclusion due to the early exploration of this direction.
> > > >
> > > > >"I have a similar concern with: "The overall correlation between cognitive test scores and guided AI performance is statistically significant (∗∗, p < 0.01).”  Is that true if you ignore Find Treasure?"
> > > >
> > > > We agree that Find Treasure is a strong factor in this as well. We will reword this into: “The correlation between cognitive test scores and guided AI performance is positive for all tasks, where Find Treasure had a strong statistical significance (∗∗, p < 0.01)."
> > > >
> > > > >"Thanks for that top25/bottom25 graph. I like that graph, I think it would make the paper stronger if you used that instead of the current Figure 9."
> > > >
> > > > Thank you for your suggestion. We will include both figures to show both partitions to provide more informative discussions on individual differences.

---

> > > > > ### Author Response · Authors · 2024-11-06
> > > > > **We would love to hear from you if you have further questions**
> > > > >
> > > > > Dear reviewer, thank you again for your thoughtful review for our manuscript. We are following up to see if you have further questions about our paper. We aim to try our best to address your concerns of our paper. Thank you again!

---

### Decision · Action_Editor_tm2b · 2024-11-21

**Recommendation:** Accept with minor revision

**Comment:**

In the discussions with reviewers, the authors have agreed to applying numerous changes, many of which are quite important. Some of these changes involve more accurate and careful claims surrounding e.g. statistical significance, and fixing cases where the incorrect figures were referenced. Therefore, I request the authors to very carefully make sure that they apply all the changes they have agreed to in their discussions with the reviewers.

Aside from that, the authors should also adjust any references to anonymity (e.g., for the github repo) and update them to no longer be anonymised for the camera-ready version.

**Audience:**

Yes.

**Claims And Evidence:**

Reviewers unanimously agree that claims and evidence are sufficiently supported (taking into account changes which the authors have agreed to apply following discussions with reviewers, but which we have not yet seen in any newly uploaded revisions). I see no reason to deviate from this assessment.

---

> ### Author Response · Authors · 2024-11-25
> **Camera-ready Submission**
>
> We would like to thank all reviewers and the action editor again for the constructive feedback. We have submitted the camera-ready version with the following revisions at the request of the AE and reviewers:
>
> 1) Reworded the claims regarding the correlation between cognitive test scores and guided AI performance.
> 2) Added top25 vs bottom25 cognitive test scorers comparison in the appendix.
> 3) Corrected typo when referring to Figures.
> 4) Updated the abstract to specify our focus on real-time decision-making tasks.
> 5) Clarifying computational resources requirement for evaluation, hyperparameter selection for the heuristic method.
> 6) Updated public project website and code URL.
>
> We have also uploaded a link to a video introduction and uploaded it as a supplementary file.